# Adjusted Net Savings of CEE and Baltic Nations in the Context of Sustainable Economic Growth: A Panel Data Analysis

**Batrancea Larissa [1], Rathnaswamy Malar Maran [2], Batrancea Ioan [2,*] , Nichita Anca [3], Rus Mircea-Iosif [4], Tulai Horia [2], Fatacean Gheorghe [2], Masca Ema Speranta [5] and Morar Ioan Dan [6]**

[1] Faculty of Business, Babes-Bolyai University, 7 Horea Street, 400174 Cluj-Napoca, Romania; lm.batrancea@tbs.ubbcluj.ro

[2] Faculty of Economics and Business Administration, Babes-Bolyai University, 58–60 Teodor Mihali Street, 400591 Cluj-Napoca, Romania; maran.rathnaswamy@econ.ubbcluj.ro (R.M.M.); horea.tulai@econ.ubbcluj.ro (T.H.); gheorghe.fatacea@econ.ubbcluj.ro (F.G.)

[3] Faculty of Economic Sciences, "1 Decembrie 1918" University of Alba Iulia, 15–17 Unirii Street, 510009 Alba Iulia, Romania; ancaramona.nichita@gmail.com

[4] National Institute for Research and Development in Constructions, Urbanism and Sustainable Spatial Development "URBAN INCERC", 117 Calea Floresti, 400524 Cluj-Napoca, Romania; mircearus2005@yahoo.com

[5] Faculty of Economics and Law, "George Emil Palade" University of Medicine, Pharmacy, Science and Technology of Targu Mures, 38 Gheorghe Marinescu Street, 540566 Targu Mures, Romania; ema.masca@umfst.ro

[6] Faculty of Economic Sciences, University of Oradea, 1 Universitatii Street, 410087 Oradea, Romania; imorar@uoradea.ro

* Correspondence: i_batrancea@yahoo.com; Tel.: +40-754-077-019; Fax: +40-264-412-570

**Abstract:** The article investigates the contribution of adjusted net savings to sustainable economic growth for 10 Central and Eastern European and Baltic nations, which are former Soviet bloc nations known as transition economies, using panel data analysis for the period 2005–2016. Our results indicated that adjusted net savings impacted on the GDP across the 10 countries analyzed. Nevertheless, national authorities are called on to implement policy changes in these countries to achieve sustainable economic growth and make an efficient transition from a brown economy towards a green economy.

**Keywords:** adjusted net savings; GDP; sustainable economic growth; brown economy; green economy

## 1. Introduction

According to the World Bank (2011), the Consumption of Fixed Capital (CFC) and the Natural Resources Depletion are deducted from the Gross National Income (GNI). Since the traditional concept of Gross Domestic Product (GDP), which generally indicates economic growth, is not adequately incorporating the sustainable development of an economy, in the 1990s the World Bank introduced Adjusted Net Savings for measuring the degree of sustainable development (Pardi et al. 2015). Sustainable development is achievable by using a hybrid sustainability model to monitor renewable natural resources within the framework of trading ecological goods and services (Chang 2011). Economic growth has been linked to $CO_2$ emissions and it considerably impacts high growth economies more than low growth economies (Aye and Edoja 2017). Carbon dioxide causes global warming and environment degradation, yet there is no evidence to suggest that economic growth has contributed to

the mitigation of $CO_2$ emissions. For this reason, we deem that it is important to consider mechanisms for regulating $CO_2$ emissions and for reducing the greenhouse gas effect.

It is a challenge to analyze the cost of degradation that triggers a loss in the GDP. Nevertheless, one of the existing methods is called Adjusted Net Savings (ANS) or "genuine savings" (McGrath et al. 2020), which manages to capture the cost of degradation concerning natural resources. Countries such as Ghana, Ecuador, Egypt or Indonesia have been implementing this method. Moreover, the World Bank (2013) also recommends ANS when evaluating economic growth and sustainable development strategies. In 2013, professionals from 35 countries attempted to develop an alternative approach, which was similar to another method implemented by the World Bank called "Wealth Accounting, and the Valuation of Ecosystem Services" (WAVES). From its onset, WAVES has become an important part of the *Global Program on Sustainability* (GPS). The GPS consists of three pillars, namely: *Pillar I*, which aims to improve global measurements regarding natural capital and ecosystem services; *Pillar II*, which informs the capacity to produce and use natural capital to support policy and planning decisions. Currently, about 20 countries worldwide implement GPS in their national policy decisions regarding natural capital, ecosystem and sustainability; *Pillar III* provides incentives to promote sustainability in the financial sector.

The recent COVID-19 pandemic has uncovered the importance of natural capital and ecosystem services. In this unprecedented context, neither WAVES nor GPS provide adequate answers to the challenges posed by global warming and environmental degradation. Natural capital is neither successfully accounted for by WAVES, nor is it accurately reflected in the level of economic growth. Therefore, the Adjusted Net Savings method attempts to reflect the factors that are responsible for economic growth and global warming.

The present article entails six parts. The first part contains the introduction, which defines the scope of the study, objectives and the unique contribution of our endeavor to future research studies within the field. The second part comprises the literature review, which discusses relevant scientific publications and reports tackling ANS and its contribution to sustainable economic growth. The third part presents the research method, model and data. The fourth part presents the results of the study. In the fifth part, the results of the study are discussed, while the sixth part presents concluding remarks and policy implications.

Our study adds to the existing literature by emphasizing possible factors that influence sustainable economic growth and global warming. Besides, it also suggests policies that could be used by nations worldwide to achieve sustainable economic growth.

## 2. Literature Review

According to the literature, Adjusted Net Savings is an adequate indicator for capturing the level of sustainable development (Boos 2015; Crabtree 2020; Everett and Wilks 1999; Jha et al. 2018; Managi and Kumar 2018), and provides authorities with information about the sustainability of national investment policies (World Bank 2018). It is designed to measure real savings in production and natural and human capital after adjusting for the depletion of natural sources (Schepelmann et al. 2010). There are multiple factors that influence a nation's level of sustainable development—proxied by Adjusted Net Savings—among which are per capita income, financial development, inflation rate, natural resource rent and banking system stability (Greasley et al. 2014; Koirala and Pradhan 2019; Ntarmah et al. 2019; Rudenko-Sudarieva and Shevchenko 2019).

The influences of carbon emissions on economic growth are highly analyzed in the economic literature. Abbasi and Riaz (2016) studied the impact of $CO_2$ emissions on the level of economic and financial development during the period 1971–2011 (divided into two stages) using vector error correction (VEC) and vector autoregression (VAR) models. According to their results, the higher level of per capita $CO_2$ emissions during the later stage of the period mentioned (i.e., 1988–2011) was generated by the development of the financial sector and liberalization. In another study, Antonakakis et al. (2017) investigated the relationship between energy consumption, carbon dioxide emissions and economic

growth on data from 106 countries during the period 1971–2011. Through a panel vector autoregression model, results revealed no evidence to suggest that renewable energy consumption increased economic growth. Namely, developed countries achieved economic growth without environmental pollution, while countries with low or zero economic growth were environmentally sustainable. Apergis and Payne (2014) rejected the previous outcomes by examining the relationship between renewable energy, real GDP per capita, carbon emissions per capita, real coal prices and real oil prices for the period 1980–2010, using data from seven Central American countries. Their results indicated the existence of a long-term relationship between the variables of interest.

A panel data analysis conducted by Kasman and Yavuz (2015) on data from new EU member states and candidate countries showed that factors such as $CO_2$ emissions, economic growth, energy consumption, trade and urbanization interact mutually in the long run. Financial development increases environmental degradation, an aspect that needs to be addressed by the national energy policies aiming to reduce carbon emissions while pursuing economic growth (Boutabba 2014). Moreover, concerning the US economy during the period 1980–2014, authors Dogan and Ozturk (2017) suggested that the American government should use more renewable sources of energy, which could trigger fewer $CO_2$ emissions.

In a study examining data from 192 countries, results showed that financial development had a substantial impact on non-renewable energy consumption, and it increased $CO_2$ emissions (Khan et al. 2020). There is no reliable accounting method that could measure natural resources depletion and the impact of environmental damages on the economy. In this context, ANS provides a viable solution to this shortcoming. A positive value of the ANS indicates sustainable economic growth for the respective country, while a negative value shows the depletion of natural resources and environmental degradation, as economic theory suggests (Blanchet et al. 2009). Sustainability has several dimensions and values, including social, economic and environmental well-being, and therefore it needs to be measured. In this regard, the literature comprises various research papers that have attempted such a measurement. Even though sustainability is measured based on the efficiency of social, environmental and economic principles, attaining sustainability is an extremely challenging a task, as public authorities such as the Federal Highway Administration within the US Department of Transportation have stated. Nevertheless, the capital approach is advocated to measure sustainable development.

Predominantly, four types of capital such as produced capital, human capital, natural capital and social capital are employed to provide goods and services for sustaining economic growth. Any change in natural capital becomes irreversible, and so it needs to be used sustainably and efficiently while achieving long-term sustainable economic growth. Natural capital differs from other types of capital in many aspects, namely replenishment and recharging. Moreover, natural capital comprises non-renewable resources such as oil, minerals, forests, fisheries and biodiversity. For this reason, it is crucial to incorporate changes regarding the management of such resources without mitigating the access of future generations. Within this context, the ANS method is considered appropriate.

Green GDP and GDP in general are deemed as measurable indicators, but the former can only be assessed via the ANS method, unlike regular GDP (Stiglitz et al. 2008; Merko et al. 2019). The ANS is considered as a proxy for sustainable development (Pardi et al. 2015). A similar idea is also advanced by the World Bank (Hartwick 1977). In the view of Pearce and Atkinson (1993), the measurement of sustainable development is challenging. Nevertheless, one can advance a "weak" sustainability indicator in terms of savings when aiming at the stability of overall capital. For that matter, using a sample of 20 countries from the Organisation for Economic Co-operation and Development (OECD), Mota and Cunha-e-Sá (2019) deemed ANS as just such an indicator, and showed that technological progress is a key factor to be considered when computing ANS.

The Paris Agreement ratified by worldwide nations underlines the importance of reducing $CO_2$ emissions and the greenhouse gas effect while achieving sustainable economic growth. It indicates the need for shifting from a brown economy to a green economy in countries across the globe. It also calls for preventing the depletion of natural resources, which should benefit future generations.

Transition economies have to develop appropriate economic policies and environmental policies, since these economies have previously functioned under the influence of centralized financial and economic systems.

## 3. Method, Model and Data

There is a relationship between sustainable economic policies and citizens' wellbeing, which ultimately conditions the prosperity of future generations. Under this framework, the concept of Adjusted Net Savings comes into play (Qasim and Grimes 2018). According to the literature, Hartwick developed the concept of reinvesting the income generated from the exploitation of non-renewable resources so that the overall natural assets could be sustained without loss. This came to be known as the Hartwick (1977, 1990).

Gnegne (2009) conducted a study across 36 nations on data from the period 1971–2000 and his results supported the existence of a positive and significant relationship between adjusted net savings and welfare. Aşıcı (2011) ran a panel data analysis including 213 countries and territories for the period 1970–2008 in order to measure economic growth and its impact on energy, net forest depletion and carbon dioxide damage. The results indicated a positive link between the variables, and emphasized that, beyond a specific level of economic growth, the phenomenon of environmental degradation would appear.

The Endogenous Growth Theory includes the Solow growth model, which suggests that productivity would increase with the help of technological advantage. Within the framework of sustainable development, the model introduced through Hartwick's Rule turned out to be relevant. According to the model, the reproducible man-made capital should substitute the depleting non-renewable resources in the production process (Dimitriou and Cook 2018; Hartwick 1977, 1990; Neumayer 2003; Solow 1974). Moreover, it helps distinguish between weak sustainability and strong sustainability.

In our panel data analysis, the following variables were considered:

GDP = Gross Domestic Product;

GNI = Gross National Income;

ASCDD = Adjusted Savings Carbon Dioxide Damage;

ASED = Adjusted Savings Energy Depletion;

ASGS = Adjusted Savings Gross Savings;

ASMD = Adjusted Savings Mineral Depletion;

ASNFD = Adjusted Savings Net Forest Depletion.

GDP is the dependent variable and the independent variables are GNI, ANSCDD, ASED, ASGS, ASMD and ASNFD.

Data were retrieved from the World Development Indicators database commissioned by the World Bank for the period 2005–2016. The software EViews version 11 was used to conduct the panel data analysis.

## 4. Results

### 4.1. Descriptive Statistics

The descriptive statistics for the variables GDP, ASCDD, ASED, ASGS, ASMD, ASNFD and GNI are presented in Table 1.

**Table 1.** Descriptive Statistics.

|  | GDP | ASCDD | ASED | ASGS | ASMD | ASNFD | GNI |
|---|---|---|---|---|---|---|---|
| Mean | 2.595000 | 1.330000 | 0.260000 | 22.30583 | 0.100833 | 0.002500 | 2.760833 |
| Median | 2.800000 | 1.100000 | 0.100000 | 22.55000 | 0.000000 | 0.000000 | 3.050000 |
| Maximum | 11.90000 | 3.000000 | 1.600000 | 29.50000 | 1.100000 | 0.100000 | 11.90000 |
| Minimum | −14.80000 | 0.500000 | 0.000000 | 10.50000 | 0.000000 | 0.000000 | −12.60000 |
| Std. Dev. | 4.641552 | 0.597417 | 0.364011 | 3.862191 | 0.245119 | 0.015678 | 4.192236 |
| Skewness | −1.228007 | 0.893172 | 2.011951 | −0.293320 | 2.599680 | 6.084870 | −0.864457 |
| Kurtosis | 6.178526 | 2.849856 | 6.479751 | 2.706432 | 8.748946 | 38.02564 | 4.906976 |
| Jarque–Bera | 80.67516 | 16.06783 | 141.5023 | 2.151642 | 300.4186 | 6874.490 | 33.12850 |
| Probability | 0.000000 | 0.000324 | 0.000000 | 0.341018 | 0.000000 | 0.000000 | 0.000000 |
| Sum | 311.4000 | 159.6000 | 31.20000 | 2676.700 | 12.10000 | 0.300000 | 331.3000 |
| Sum Sq. Dev. | 2563.737 | 42.47200 | 15.76800 | 1775.066 | 7.149917 | 0.029250 | 2091.406 |
| Observations | 120 | 120 | 120 | 120 | 120 | 120 | 120 |

Source: Our computations.

As can be seen from Table 1, the mean values for the variables ASCDD, ASED, ASMD are higher than the one for ASNFD. Except for ASNFD, the skewness of all the other six variables is below 3, and therefore the model can be deemed as valid. The higher values registered by the kurtosis corresponding to all variables of interest indicate that the distribution of data is leptokurtic. Regarding the Jarque–Bera test, with the exception of ASGS, the *p*-value for all the other variables is below 5%, indicating that the model is significant.

### 4.2. Analysis of the Correlations between Variables

Based on Table 2, it can be observed that the variables of interest, both independent and dependent, registered very low or no correlations between them. Consequently, the issue of multicollinearity can be ruled out from our analyses.

**Table 2.** Correlation Matrix.

|  | GDP | ASCDD | ASED | ASGS | ASMD | ASNFD | GNI |
|---|---|---|---|---|---|---|---|
| GDP | 1 |  |  |  |  |  |  |
| ASCDD | 0.1448 | 1 |  |  |  |  |  |
| ASED | −0.0472 | 0.2312 | 1 |  |  |  |  |
| ASGS | −0.0365 | −0.0022 | 0.0064 | 1 |  |  |  |
| ASMD | 0.0548 | 0.6149 | −0.0702 | −0.2870 | 1 |  |  |
| ASNFD | 0.0255 | 0.1354 | −0.0706 | −0.1057 | 0.0650 | 1 |  |
| GNI | 0.9181 | 0.1086 | 0.0387 | 0.0167 | 0.0181 | 0.0091 | 1 |

Source: Our computations.

### 4.3. Unit Root Test and Hadri Test

In panel data analysis, unit root tests aim to investigate data quality and provide an accurate description of the economic phenomenon (Dickey 1976; Dickey and Fuller (1979, 1981); Fountis and Dickey 1989; Levin and Lin 1993; Im et al. 2003; Maddala and Wu 1999; MacKinnon (1991, 1996); Maddala and Kim 1998; Philips and Xiao 1998; Stock 1994). In our case,

MacKinnon critical values are used in EViews to examine the hypothesis of the unit root test. The Hadri test for the unit root is conducted and the hypothesis is built around stationarity.

$$\Delta Y_t = \beta y_{t-1} + \varepsilon \tag{1}$$

$$\Delta Y_t = b_0 + \beta y_{t-1} + \varepsilon \tag{2}$$

$$\Delta Y_t = b_0 + \beta y_{t-1} + b_2 + \varepsilon \tag{3}$$

The null hypothesis is the following:

$$H_0 : \beta = 0 \tag{4}$$

The alternate hypothesis is the following:

$$H_a : \beta < 0 \tag{5}$$

The null hypothesis of the unit root test is rejected at level and at first difference according to the Levin, Lin and Chu $t$, since $p$-values are below 0.05. The null hypothesis is also rejected at level and at first difference in the cases of the Im, Pesaran and Shin $W$-statistics, ADF-Fisher $chi$-square and PP-Fisher $chi$-square, as shown in Table 3.

The null hypothesis of stationarity is rejected by the Hadri test at first difference, but not at level.

Among the 10 nations considered, Poland has the lowest HAC variance at level and at first difference, with the values 2.531875 and 0.562562, therefore it can easily frame policies to achieve sustainable economic growth, and reduce $CO_2$ emissions and the greenhouse gas effect. This indicates a considerable performance in managing the low carbon economy with adjusted net savings. Estonia holds the highest HAC variance with 50.0116 at level, but at first difference this is not the highest value. Namely, with better financial management of the adjusted net savings, the country could design economic and environmental policies to mitigate $CO_2$ emissions and the greenhouse effect while achieving sustainable economic growth. Hungary has the second-lowest variance both at level and at first difference, with 10.29513 and 2.648750, which indicates consistency in promoting sustainable economic growth. Nevertheless, the country must implement suitable policy changes to mitigate pollution. Lithuania is third in this ranking with 29.40377 and 7.879279 of HAC variance at level and at first difference, meaning that this country can easily reform its policies to achieve sustainable economic growth, employing adjusted net savings.

Romania has obtained the fourth-highest HAC variance at level and at first difference (i.e., 22.78972 and 4.168208), which calls for immediate policy changes to achieve sustainable economic growth. The existence of inconsistent policies is reflected in the Latvian data because the HAC variance here has reached the highest value at level and at first difference (i.e., 49.32521 and 19.56847). Starting from these results, we believe that authorities in Latvia are called on to introduce reforms in their economic and environmental policies that should be regularly reviewed. Slovakia ranked seventh at level with a value of 15.92076 and sixth at first difference with 2.880975. Slovenia is sixth at level with 15.47963 and has reached a value of 4.77524 at first difference. Authorities in Slovakia have taken greater steps to improve sustainable economic growth, yet the country still needs to consistently reform its economic and environmental policies. On the other hand, Slovenia has to take drastic measures to achieve the same goals.

Bulgaria and Czech Republic, which ranked 9th and 10th, have reached HAC variance at level and at first difference as follows: Bulgaria has registered the values of 14.76539 and 3.042975, while the Czech Republic 14.262224 and 2.721938. According to these results, authorities in Bulgaria should revise economic and environmental policies as compared to Czech authorities to ensure sustainable economic growth in the long run.

**Table 3.** Unit Root Test and Hadri Test.

| | **Panel Unit Root Test: Summary Series: GDP** | | | | **Panel Unit Root Test: Summary Series: D(GDP)** | | | |
|---|---|---|---|---|---|---|---|---|
| Method | Statistic | Prob. ** | Cross-Sections | Obs. | Statistic | Prob. ** | Cross-Sections | Obs. |
| Null: Unit root (assumes common unit root process) | | | | | | | | |
| Levin, Lin and Chu t * | −7.53233 | 0.0000 | 10 | 107 | −11.4727 | 0.0000 | 10 | 94 |
| Null: Unit root (assumes individual unit root process) | | | | | | | | |
| Im, Pesaran and Shin W test | −3.63088 | 0.0001 | 10 | 107 | −6.28179 | 0.0000 | 10 | 94 |
| ADF-Fisher chi-square | 45.3809 | 0.0010 | 10 | 107 | 75.8515 | 0.0000 | 10 | 94 |
| PP-Fisher chi-square | 36.4986 | 0.0134 | 10 | 110 | 102.528 | 0.0000 | 10 | 100 |
| | **Null Hypothesis: Stationarity Series: GDP** | | | | **Null Hypothesis: Stationarity Series: D(GDP)** | | | |
| Method | | Statistic | Prob. ** | | | | Statistic | Prob. ** |
| Hadri Z-stat | | 0.63450 | 0.2629 | | | | 5.23213 | 0.0000 |
| Heteroscedastic consistent Z-stat | | 1.39141 | 0.0821 | | | | 5.86540 | 0.0000 |
| | Intermediate results on GDP | | | | Intermediate results on D(GDP) | | | |
| Cross section | LM | Variance HAC | Bandwidth | Obs. | LM | Variance HAC | Bandwidth | Obs. |
| Romania | 0.3087 | 22.78972 | 1.0 | 12 | 0.3518 | 4.168208 | 7.0 | 11 |
| Bulgaria | 0.2641 | 14.76539 | 1.0 | 12 | 0.5000 | 3.042975 | 10.0 | 11 |
| Hungary | 0.1459 | 10.29513 | 1.0 | 12 | 0.5000 | 2.648760 | 10.0 | 11 |
| Poland | 0.4487 | 2.531875 | 0.0 | 12 | 0.4545 | 0.562562 | 9.0 | 11 |
| Czech Republic | 0.2014 | 14.26224 | 1.0 | 12 | 0.5000 | 2.721938 | 10.0 | 11 |
| Slovenia | 0.1729 | 15.47963 | 1.0 | 12 | 0.3582 | 4.775224 | 7.0 | 11 |
| Slovak | 0.3100 | 15.92076 | 0.0 | 12 | 0.4545 | 2.880975 | 9.0 | 11 |
| Estonia | 0.1146 | 50.01160 | 2.0 | 12 | 0.5000 | 10.80234 | 10.0 | 11 |
| Latvia | 0.1969 | 49.32521 | 0.0 | 12 | 0.3126 | 19.56847 | 6.0 | 11 |
| Lithuania | 0.1594 | 29.40377 | 3.0 | 12 | 0.5000 | 7.879279 | 10.0 | 11 |

* Note: High autocorrelation leads to the severe size distortion in Hadri test, leading to the over-rejection of the null hypothesis. ** Probabilities are computed assuming asymptotic normality. Source: Our computations.

### 4.4. Pooled OLS, Fixed Effect, Random Effect, Hausman Test and Fully Modified Least Squares (FMOLS)

The random effect model is considered when the hull hypothesis of the Hausman test is accepted. If the null hypothesis is rejected, the fixed effect model is taken into consideration. Moreover, FMOLS indicates the contributions of each of the variables of interest to achieving sustainable economic growth.

$$Y_{it} = \alpha + X_{it}\beta + \delta_i + \gamma_t + \varepsilon_{it} \tag{6}$$

$$Y_{it} = \alpha + X_{it}\beta_i + \delta_i + \gamma_t + \varepsilon_{it} \tag{7}$$

$$Y_{it} = \alpha + X_{it}\beta_t + \delta_i + \gamma_t + \varepsilon_{it} \tag{8}$$

$$GDP_{it} = \beta_1 + \beta_2 ASCDD + \beta_3 ASED + \beta_4 ASGS + \beta_5 ASMD$$
$$+ \beta_6 ASNFD + \beta_7 GNI + \epsilon_{it} \tag{9}$$

FMOLS examines the long-run relationship among variables (Kao and Chiang 2001). After investigating the properties of the OLS estimator, it was found that the bias-corrected OLS model did not generally improve the OLS estimator and, therefore, the FMOLS estimator was suggested as being adequate (Saikkonen 1991; Stock and Watson 1993).

$$y_t = Ax_t + u_a$$

Table 4 displays the results of the pooled OLS, fixed effect, random effect, Hausman test and FMOLS.

As can be seen from Table 4, in terms of the random effect model, the factors ASCDD and GNI positively influenced the variable GDP, while the factor ASED triggered a decrease in the level of economic growth.

As indicated by the results of the FMOLS model, all variables impacted economic growth. Thereby, the results of the FMOLS model support the idea that adjusting net savings helps measure sustainable economic growth within an economy. Therefore, it is possible to achieve an adequate growth of human capital and natural assets without a considerable depletion. Consequently, phenomena such as environmental degradation and global warming can be restricted accordingly.

**Table 4.** Pooled OLS, Fixed effect, Random effect and Hausman Test.

| | **Pooled OLS Model** Dependent Variable: GDP | | | | **Fixed Effect Model** Dependent Variable: GDP | | | |
|---|---|---|---|---|---|---|---|---|
| **Variable** | **Coefficient** | **Std. Error** | **t-Statistic** | **Prob.** | **Coefficient** | **Std. Error** | **T-statistic** | **Prob.** |
| C | 0.986365 | 1.052900 | 0.936808 | 0.3509 | −1.042355 | 2.205499 | −0.472616 | 0.6375 |
| ASCDD | 0.810979 | 0.394260 | 2.056964 | 0.0420 | 2.132852 | 0.879952 | 2.423828 | 0.0171 |
| ASED | −1.409720 | 0.490239 | −2.875578 | 0.0048 | 0.310946 | 0.899573 | 0.345659 | 0.7303 |
| ASGS | −0.080192 | 0.046380 | −1.729024 | 0.0865 | −0.100486 | 0.066597 | −1.508862 | 0.1344 |
| ASMD | −0.990257 | 0.962121 | −1.029243 | 0.3056 | 1.718782 | 2.544392 | 0.675518 | 0.5008 |
| ASNFD | −2.488708 | 10.76673 | −0.231148 | 0.8176 | 3.104594 | 12.14008 | 0.255731 | 0.7987 |
| GNI | 1.011067 | 0.039546 | 25.56678 | 0.0000 | 1.006999 | 0.041692 | 24.15353 | 0.0000 |
| R-squared | 0.858105 | F-statistic | 113.8943 | | R-squared | 0.872506 | F-statistic | 47.44831 |
| Adjusted R-squared | 0.850571 | Prob. (F-statistic) | 0.000000 | | Adjusted R-squared | 0.854117 | Prob. (F-statistic) | 0.000000 |
| | **Random Effect Model** Dependent Variable: GDP | | | | **Correlated Random Effects–Hausman Test** | | | |
| **Variable** | **Coefficient** | **Std. error** | **t-statistic** | **Prob.** | **Test Summary** | **Chi-Sq. Statistic** | **Chi-Sq. d.f.** | **Prob.** |
| | | | | | Cross-section random | 11.486496 | 6 | 0.0745 |
| | | | | | Cross-section random effects test equation. Dependent variable: GDP; Method: Panel Least Squares Sample: 2005–2016; Periods included: 12 Cross-sections included: 10 Total panel (balanced) observations: 120 | | | |

**Table 4.** *Cont.*

| | | | | | | | | |
|---|---|---|---|---|---|---|---|---|
| C | 0.986365 | 1.040331 | 0.948127 | 0.3451 | −1.042355 | 2.205499 | −0.472616 | 0.6375 |
| ASCDD | 0.810979 | 0.389554 | 2.081815 | 0.0396 | 2.132852 | 0.879952 | 2.423828 | 0.0171 |
| ASED | −1.409720 | 0.484387 | −2.910320 | 0.0043 | 0.310946 | 0.899573 | 0.345659 | 0.7303 |
| ASGS | −0.080192 | 0.045826 | −1.749914 | 0.0828 | −0.100486 | 0.066597 | −1.508862 | 0.1344 |
| ASMD | −0.990257 | 0.950636 | −1.041678 | 0.2998 | 1.718782 | 2.544392 | 0.675518 | 0.5008 |
| ASNFD | −2.488708 | 10.63820 | −0.233941 | 0.8155 | 3.104594 | 12.14008 | 0.255731 | 0.7987 |
| GNI | 1.011067 | 0.039074 | 25.87567 | 0.0000 | 1.006999 | 0.041692 | 24.15353 | 0.0000 |

| | | | |
|---|---|---|---|
| R-squared | 0.858105 | F-statistic | 113.8943 |
| Adjusted R-squared | 0.850571 | Prob. (F-statistic) | 0.000000 |

**Dependent variable: GDP**
**Method: Panel Fully Modified Least Squares (FMOLS)**

| Variable | Coefficient | Std. error | t-statistic | Prob. | R-squared | −86.850699 |
|---|---|---|---|---|---|---|
| ASCDD | 80.91177 | 21.39724 | 3.781412 | 0.0129 | Adjusted R-squared | −174.701397 |
| ASED | 436.7762 | 59.23867 | 7.373160 | 0.0007 | | |
| ASGS | −9.870857 | 1.813296 | −5.443599 | 0.0028 | | |
| ASMD | −379.9181 | 75.26561 | −5.047699 | 0.0039 | | |
| ASNFD | 467.9440 | 93.47772 | 5.005941 | 0.0041 | | |
| GNI | 11.90856 | 2.650747 | 4.492529 | 0.0064 | | |

Source: Our computations.

## 4.5. Vector Autoregression Estimates and VAR

VAR is one of the most popular models used in panel data analysis of the economic system. $T$ is considered as the number of time series for $N$ groups, and it estimates $N$ distinct regressions for computing the mean in this model (Pesaran et al. 1999). To estimate the model, we used data divided into time periods, $t = 1, \ldots, T$, and several groups, $I = 1.2, \ldots, N$, as given by the VAR model presented below:

$$y_t = A_1 y_{t-1} + A_2 y_{t-2} + \ldots + A_p y_{t-P} + B_{x_t} + \mu_t \tag{10}$$

$$t = 1, 2, \ldots, T \tag{11}$$

$$y_t = A_1 y_{t-1} + A_2 y_{t-2} + \ldots + A_p y_{t-P} + \mu_t$$

$$t = 1, 2, \ldots, T \tag{12}$$

$$\Delta y_t = \Pi\, y_{t-1} + \sum_{i=1}^{P-1} \Gamma_i \Delta y_{t-i} + \mu_t$$

where

$$\Pi = \sum_{i=1}^{P} A_i - I$$

$$\Gamma_i = \sum_{j=i+1}^{P} A_j$$

The formula for the cointegration relationship is written as follows:

$$y_t = \alpha\beta' y_{t-1} + \sum_{i=1}^{P-1} \Gamma_i \Delta y_{t-i} + \mu_t \tag{13}$$

In terms of the vector autoregression estimates shown in Table 5, the R-squared values for ASCDD, ASED, ASGS, ASMD and ASNFD indicated the high efficiency of these variables. Contrariwise, the R-squared of GNI was below 0.5.

**Table 5.** Vector Autoregression Estimates.

|  | GDP | ASCDD | ASED | ASGS | ASMD | ASNFD | GNI |
|---|---|---|---|---|---|---|---|
| GDP (−1) | 0.330601 | −0.024555 | 0.026471 | 0.029958 | 0.001406 | −0.000233 | 0.530035 |
|  | (0.25889) | (0.01065) | (0.01049) | (0.13083) | (0.00367) | (0.00072) | (0.22102) |
|  | [1.27700] | [−2.30629] | [2.52368] | [0.22899] | [0.38282] | [−0.32425] | [2.39816] |
| GDP (−2) | −0.753851 | −0.001868 | 0.002335 | 0.110105 | −0.005562 | −7.01 | −0.490022 |
|  | (0.24527) | (0.01009) | (0.00994) | (0.12395) | (0.00348) | (0.00068) | (0.20939) |
|  | [−3.07351] | [−0.18523] | [0.23494] | [0.88832] | [−1.59804] | [−0.10278] | [−2.34019] |
| C | 3.838509 | −0.254549 | 0.132283 | 4.656892 | −0.001241 | 0.008681 | 4.047393 |
|  | (2.58679) | (0.10638) | (0.10480) | (1.30721) | (0.03671) | (0.00719) | (2.20838) |
|  | [1.48389] | [−2.39277] | [1.26219] | [3.56248] | [−0.03382] | [1.20661] | [1.83274] |
| R-squared | 0.389363 | 0.936236 | 0.845015 | 0.756915 | 0.956780 | 0.673096 | 0.435960 |
| Adj. R-squared | 0.288787 | 0.925734 | 0.819488 | 0.716877 | 0.949661 | 0.619253 | 0.343059 |

Source: Our computations.

The results from Table 6 explain the nature of the long-term relationship between the variables considered, which influence economic growth.

**Table 6.** Estimation Method: Least Squares.

| **Estimation Method: Least Squares** <br> **Sample: 2007–2016; Included Observations: 100; Total System (Balanced) Observations: 700** | | | | |
|---|---|---|---|---|
|  | **Coefficient** | **Std. Error** | **t-Statistic** | **Prob.** |
| C (1) * ASCDD | 0.330601 | 0.258889 | 1.277002 | 0.2021 |
| C (2) * ASED | −0.753851 | 0.245274 | −3.073506 | 0.0022 |
| C (3) * ASMD | 1.813993 | 2.730482 | 0.664349 | 0.5067 |
| C (4) * ASNFD | 0.816012 | 2.902347 | 0.281156 | 0.7787 |
| C (5) * GNI | −8.752199 | 2.775356 | −3.153541 | 0.0017 |
| R-squared | 0.389363 |  |  |  |
| Adjusted R-squared | 0.288787 |  |  |  |

Source: Our computations.

### 4.6. The Wald Test

Furthermore, we conducted the Wald test to analyze the short-term relationship between the variables of interest.

In Table 7, one can see the results of the Wald tests conducted for our variables. It can be concluded that the variables GDP, ASED, ASMD, ASNFD and GNI establish a short-term equilibrium between them, while the variables ASCDD and ASGS do not establish such a relationship. Consequently, it could be stated that these variables are not sufficient to compensate for the consumption of natural resources in the short run. Moreover, in the long run, among the corrective measures that can be enacted are the attempt to increase natural capital, or to use technology that mitigates $CO_2$ emissions, such that total natural assets are supported without environmental degradation.

**Table 7.** Wald Test.

| **Wald Test I: GDP** | | | |
|---|---|---|---|
| **Test Statistic** | **Value** | **d.f.** | **Probability** |
| Chi-square | 15.75434 | 6 | 0.0151 |
| Null Hypothesis: C (3) = C (5) = C (7) = C (9) = C (11) = C (13) = 0 | | | |
| **Wald Test II: ASCDD** | | | |
| Chi-square | 2.071322 | 6 | 0.9130 |
| Null Hypothesis: C (19) = C (21) = C (23) = C (25) = C (27) = C (29) = 0 | | | |
| **Wald Test III: ASED** | | | |
| Chi-square | 77.50685 | 6 | 0.0000 |
| Null Hypothesis: C (33) = C (35) = C (37) = C (39) = C (41) = C (43) = 0 | | | |
| **Wald Test IV: ASGS** | | | |
| Chi-square | 8.558206 | 6 | 0.2000 |
| Null Hypothesis: C (49) = C (51) = C (53) = C (55) = C (57) = C (59) = 0 | | | |
| **Wald Test V: ASMD** | | | |
| Chi-square | 118.1690 | 6 | 0.0000 |
| Null Hypothesis: C (63) = C (65) = C (67) = C (69) = C (71) = C (73) = 0 | | | |
| **Wald Test VI: ASNFD** | | | |
| Chi-square | 81.82548 | 6 | 0.0000 |
| Null Hypothesis: C (78) = C (80) = C (82) = C (84) = C (86) = C (88) = 0 | | | |
| **Wald Test VII: GNI** | | | |
| Chi-square | 19.43980 | 6 | 0.0035 |
| Null Hypothesis: C (93) = C (95) = C (97) = C (99) = C (101) = C (103) =0 | | | |

Source: Our computations.

## 5. Discussion

Environmental policies must be designed suitably to support economic growth in the long run. Case in point, in 2007 the UK enacted economic policies focused on the environment to save annually GBP 6.4 billion. Namely, it adopted no- or low-cost measures to enhance energy efficiency (Pearson 2003).

In general, the process of decoupling production from environmental damages has two elements. The first element refers to relative decoupling, which may indicate higher environmental damage relative to the GDP growth. The second element is absolute and may indicate a reduction in the environmental damage while the GDP increases.

In Table 8, shading represents the impact of economic growth on $CO_2$ emissions and global warming. As one can observe, Ireland achieved the highest GDP, at 258, with 126 $CO_2$ and 106 particulates. Turkey holds the second-highest GDP of 173, with 184 $CO_2$, 166 NOx and 128 SOx. Compared to Ireland, Turkey has incurred the maximum environmental damage. It can be deemed that the country has not achieved an adequate level of Adjusted Net Savings, which might counterbalance the mitigation of natural capital, which results in the depletion of natural resources on the one hand, and in higher environmental degradation on the other hand. Should this trend continue, it might be possible that future generations experience a scarcity of natural resources.

**Table 8.** GDP and domestically produced emission indices for selected OECD countries.

| | GDP | Sox | NOx | Particulates | CO | VOC | CO$_2$ |
|---|---|---|---|---|---|---|---|
| France | 132 | 35 | 66 | 67 | 50 | 52 | 98 |
| Germany | 123 | 10 | 50 | 10 | 33 | 35 | 82 |
| Ireland | 258 | 38 | 95 | 106 | 55 | 58 | 126 |
| Japan | 120 | 76 | 94 | | 67 | 88 | 107 |
| Portugal | 135 | 69 | 104 | 133 | 70 | 94 | 143 |
| Turkey | 173 | 128 | 166 | | 92 | | 184 |
| UK | 143 | 19 | 55 | 53 | 29 | 41 | 85 |
| USA | 155 | 63 | 74 | 81 | 62 | 69 | 116 |

Source: Calculations based on OECD (2007) and Everett et al. (2010).

The USA is the third country to achieve a higher GDP of 155, with 116 $CO_2$. Among Ireland, Turkey and the USA, the latter has achieved a higher GDP with lower environmental damage. The UK is the fourth nation to achieve a high GDP with 85 $CO_2$, that is, a minimum environmental damage. This indicates that a higher level of economic growth does not necessarily result in higher $CO_2$ emissions as long as Adjusted Net Savings adequately compensate for the consumption of natural resources. Germany registered a GDP of 123, with the lowest $CO_2$ emissions (i.e., 82) and particulates. Moreover, Japan obtained a GDP of 120 with $CO_2$ emissions of 107. When comparing Germany to Japan, it could be stated that an increase in economic growth does not trigger a proportional increase in the degree of environmental damage. In other words, the degree of environmental damage can be lessened through higher Adjusted Net Savings (namely, ASCDD, ASED, ASMD, ASNFD).

From Table 8, the following conclusions arise:

1. Generally, there seems to be a direct link between economic growth and environmental damage, but not for all countries;
2. Economic growth can be achieved without environmental damage when Adjusted Net Savings are sufficient in order to compensate for the consumption of natural resources;
3. Not every increase in environmental damage leads to economic growth;
4. The relationship between economic growth and environmental damage is less obvious.

The literature reports that economic growth—especially in the CEE and Baltic nations—along with other factors such as corruption and political instability contribute to the depletion of natural resources without adequately compensating for the consumption of natural resources (Zugravu et al. 2008). It also argues that trade openness promotes environmental quality (Dean 1992, 2002). Moreover, green growth triggers economic growth, and it is environmentally sustainable because it reduces pollution and greenhouse gas emissions. For these reasons, similar strategies can be implemented for minimizing waste and preventing the inefficient use of natural resources (OECD 2015).

The 10 CEE and Baltic nations in our sample ranked differently in terms of sustainable economic growth, as shown by the results of the Hadri tests. Hence, in Estonia, economic and environmental policies should facilitate sustainable economic growth, along with the reduction of $CO_2$ emissions and the greenhouse gas effect. Despite the fact that the country has achieved the highest HAC variance at level, it has also registered the second lowest value at first difference. In other words, this reflects that the level of Adjusted Net Savings is not adequate in the long run. Poland has the lowest HAC variance at level and this trend continues at first difference.

The FMOLS results from Table 4 support the abovementioned observations, as all variables of interest contributed to sustainable economic growth proxied by GDP. Namely, in the long run, ASCDD increased the GDP by 80.9%. Under the influence of the variable ASED, national GDP was augmented by 436.7%. This result suggests that countries could improve energy production by using modern technology, namely renewable energy sources that mitigate the levels of pollution and $CO_2$ emissions. The variable ASMD increased the GDP by 379.9%, thus indicating that environmental degradation could be regulated with appropriate policies. Since ASNFD increased the GDP by 467.9%, nations could enact appropriate policies to protect forests and prevent massive deforestation. Besides, we noticed that GNI and ASGS significantly influenced the GDP by 11.9% and 9.8%, respectively.

## 6. Conclusions and Policy Implications

The Adjusted Net Savings method developed by the World Bank in the 1990s remains efficient in the process of ensuring sustainable economic growth. Nevertheless, the literature emphasizes that creating a green economy is more difficult than the method suggests.

The phenomenon of climate change (Fankhauser 1995) has become more and more present in modern life, and it ought to be solved. For this purpose, governments worldwide came up with the Paris Agreement, a landmark cooperative international treaty set out to reduce $CO_2$ emissions. Nevertheless, the USA, one of the largest polluters worldwide, withdrew from the commitments of the Paris Agreement.

It is generally accepted that carbon dioxide emissions impose higher costs on economic growth. In addition, poverty reduction represents a global challenge, and is one of the millennium and sustainable development goals (Eisenmenger et al. 2020; World Bank 2007). In this context, economic development without environmental degradation is a constant challenge for all countries. The aim of achieving a low carbon economy or a green economy fits perfectly with the concept of sustainable development. We believe that adjusted net savings constitute one of the means to attain this aim. Therefore, enabling adequate conditions to shift from a brown economy to a green economy remains a challenge despite the concerted efforts of world nations (Batrancea et al. 2020a, 2020b; UNEP 2011), including the following: carbon pricing; adequate fiscal policy and taxation (Batrancea et al. 2012, 2018, 2019; Kogler et al. 2013; Nichita et al. 2019; Roux Valentini Coelho Cesar et al. 2019); carbon trading system.

A green economy postulates economic growth without environmental degradation, and it facilitates poverty eradication. In the same line of thought, sustainable economic growth reduces $CO_2$ emissions and increases the level of employment.

Each nation should learn to place the wellbeing of its citizens at the center of its policymaking. Various challenges like the COVID-19 pandemic might appear and impede authorities' commitment to shift from a brown economy to a low carbon or no carbon economy (World Bank 2020). According to

the UNEP (2011), providing sufficient financial resources to achieve such an economy raises challenges regarding the fiscal policy for both developed and developing economies, including the CEE and Baltic nations. As the World Bank (2011) puts it, a sustainable economy does not aim at the same level of citizens' wellbeing worldwide, because there are differences in cultural capital among nations, which change depending on the level of community development and living standards. This reality should be taken into account while designing public policies in the pursuit of sustainable economic growth for both CEE and Baltic nations, especially because there seems to be a direct link between adjusted net savings and sustainable economic growth (World Bank 2013).

The method based on adjusted net savings informs the process of achieving sustainable economic growth through monitoring the adequacy of compensating the depletion of natural capital (including minerals and forests) with investments in other assets, like human capital and infrastructure.

Public authorities should design appropriate policies to successfully implement the method of adjusted net savings for each type of capital. Moreover, such national policies should aim at consolidating natural resources more than at encouraging their massive consumption or depletion.

All in all, policymakers must enact new policies in accordance with the other elements of adjusted net savings, for the purpose of increasing the GDP, consolidating a strong level of sustainable economic growth, and reducing $CO_2$ emissions and the greenhouse gas effect.

**Author Contributions:** Conceptualization, B.I.; methodology, B.I.; software, M.I.D.; validation, R.M.M.; formal analysis, R.M.M.; resources, F.G.; data curation, M.E.S.; writing—original draft preparation, R.M.M.; writing—review and editing, B.L. and N.A.; visualization, T.H.; supervision, R.M.-I.; project administration, B.I. All authors have read and agreed to the published version of the manuscript.

**Funding:** This research was funded by the Babes-Bolyai University of Cluj-Napoca through the Grants for Supporting Employees' Competitiveness AGC–30121/17.01.2020.

**Acknowledgments:** The authors gratefully thank the editors and the anonymous reviewers of the journal for their useful and constructive comments, which improved the quality of the paper.

**Conflicts of Interest:** The authors declare no conflict of interest.

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
