# Peer review of "Adjusted Net Savings of CEE and Baltic Nations in the Context of Sustainable Economic Growth: A Panel Data Analysis"

_jrfm, doi:10.3390/jrfm13100234_

Round 1
Reviewer 1 Report
The paper "Adjusted Net Savings of CEE and Baltic Nations in Sustainable Economic Growth: A Panel Data Analysis" covers an interesting research topic. It has the usual structure: Introduction, literature review, variables and Methods, Results, discussion and conclusions. The literature review is enough, and the methods seem to me adequate to the ends of the article. But it has some drawbacks: the quality of the English language is not enough since there are some errors, omissions and other aspects that need to be corrected. A deep review made by a native English speaker. The description of the variables and even the models used are confused or not clear. Even the results are not all well interpreted, sometimes they tell the contrary of what is written by the authors. My suggestion is (i) not to accept the article as it is, so (ii) review it carefully and (iii) resubmit it again. I can review a new version of the paper if it is needed.
Author Response
Reply to Reviewer 1
Dear Colleague,
We are extremely grateful for your recommendation and reviewing of our article Adjusted Net Savings of CEE and Baltic Nations in Sustainable Economic Growth: A Panel Data Analysis, (jrfm-897856). We have included your recommendations and we address them in a point-by-point manner as follows.
Point 1: The paper "Adjusted Net Savings of CEE and Baltic Nations in Sustainable Economic Growth: A Panel Data Analysis" covers an interesting research topic.
Reply to point 1: We are grateful for your kind words and appreciation of our research efforts in finding an interesting topic. We are motivated and committed to carry out similar research publications in future, too.
Point 2: It has the usual structure: Introduction, literature review, variables and Methods, Results, discussion and conclusions.
Reply to point 2: Thank you for your observation. We have improved the structure of the paper, starting from the: abstract; introduction; literature review; method, model, and data; results; discussion; conclusions; policy implications.
Point 3: The quality of the English language is not enough since there are some errors, omissions and other aspects that need to be corrected. A deep review made by a native English speaker.
Reply to point 3: Thank you for your observation. We have carefully proofread the article and amended the English language errors, thus substantially improving the quality of the message and the level of understanding for the journal readers.
Point 4: The description of the variables and even the models used are confused or not clear. Even the results are not all well interpreted, sometimes they tell the contrary of what is written by the authors.
Reply to point 4: Thank you for your observation. We have clarified the description of the variables of interest, the models and the interpretation of the results. Please find all these changes in the revised manuscript.

Reviewer 2 Report
See in attachment

Author Response
Reply to Reviewer 2
Dear Colleague,
We are extremely grateful for your valuable comments and recommendations that helped improve the quality of our research article. We have addressed your comments in a point-by-point manner, please find our replied below.
Point 1: The subject in this manuscript seems not clear, so it can't attract reader’s interesting. I hope authors to modify it become have a clear and attracting subject.
Reply to point 1: We have improved the clarity and readability of our article, following your recommendations. Please find these changes in the revised manuscript.
Point 2: The abstract need to rewrite to become informative. Please try to highlight in what already known and what gap in the literature and what authors have added in this study.
Reply to point 2: Thank you for your suggestion. We have revised and rewritten the article abstract in order to make it more informative and clear.
Point 3: In introduction section, it’s not enough to state the current works. I suggest authors could revise this section and state the current work in detail, including the motivation, the main difficulties, the methods and the improvements compared with previous works.
Reply to point 3: Thank you very much for your observation. We have improved and extended the introduction, please find these changes in the revised manuscript.
Point 4: In section 3. Method and Results, authors can use a Table to summary and compare the result difference between original Adjusted Net Saving (ANS) and the new variables (line 160) that authors take into accounting.
Reply to point 4: We have revised Section 3 containing the results, please find the changes in the amended manuscript.
Point 5: The indicators such as CO2, NOx in Table 8, why not use indictors: ASCDD, ASED, ASGS, ASMD or GNS, CFC, EDU, NRD, GHG, POL. What relationship between Table 8 and Adjusted Net Saving?
Reply to point 5: We have used Table 8 in order to indicate the relationship between CO2 emissions and GDP for various OECD member states. It emphasizes the efforts made by these nations to regulate CO2 emissions for achieving sustainable economic growth. This is the specific purpose for including the table in the discussion section.
Point 6: What have been added in this study authors need to point out.
Reply to point 6: We have mentioned the literature gap that our article fills in the revised version.
Point 7: I hope authors discuss about constraint and suggestions for future research.
Reply for point 7: Thank you for your suggestion. We have addressed these aspects in sections 6 and 7 regarding the concluding remarks and policy implications.

Round 2
Reviewer 1 Report
In my point of view the paper was partly improved but it is not enough. The renewed version has not yet the quality needed to be published since it needs many changes. The literature review can also be improved (enlarged). My recommendations are to review carefully sections 3. methods and data description, section 4. presentation of the results and 5. Interpretations of the results. The last point (section 7. Policy implications) of the article needs not to be isolated in an independent section but can be included in the discussion one or even in the conclusions.
Other aspects: the numbering of the mathematical expressions of the econometric models should be sequential beginning in (1), (2)... The authors cannot repeat the same number with a different expression (for instance in 2.3.3). Not all the expressions are numbered. Several expressions with different numbers are exactly the same. There are expressions that are not needed. There are models that are not previously introduced and explained (f. ex, the one of the table 6.).
The interpretation of the results is very poor and have many limitations. Some interpretations of the descriptive statistics are not needed and some of them are not well defined.
There are still some errors and omissions in the English language.
We could continue to list other points of the paper with negative comments, but we think it is enough for now.
Author Response
Reply to Reviewer 1, Second Round
Dear Colleague,
We are grateful for all your valuable recommendations and suggestions provided to improve the quality of this article. Please find below our reply in a point-by-point manner.
Point 1: The literature review can also be improved (enlarged).
Reply to Point 1: Thank you very much for your suggestion. We have improved the literature and included the following relevant sources, as given below:
- Aşıcı, Ahmet Atıl. 2011. Economic growth and its Impact on Environment: A panel data analysis. Ecological Indicators 24: 324–333. doi: 10.1016/j.ecolind.2012.06.019.
- Everett, Glyn, and Alex Wilks. 1999. The World Bank’s genuine savings indicator: A useful measure of sustainability? Bretton Woods Project.
- Gnegne, Yacouba. 2009. Adjusted net saving and welfare change. Ecological Economics 68: 1127–39.
- World Bank. 2020. The Global Economic Outlook during the COVID-19 Pandemic: A Changed World. Washington, DC: World Bank.
- Schepelmann, Philipp, Yanne Goossenes, Artuu Makipaa, Martin Herrndorf, Verena Klees, Michael Kuhndt, and Esabel Sand (Eds.). 2010. Towards Sustainable Development: Alternatives to GDP for Measuring Progress. Wuppertal Institute for Climate, Environment and Energy: Wuppertal.
Point 2: To review carefully sections 3. methods and data description, section 4. presentation of the results and 5. Interpretation of the results.
Reply to Point 2: Thank you very much for your suggestion. We have amended the sections you indicated by carefully rewording them and reinterpreting the results. Please find these changes in the revised version of the manuscript.
Point 3: The last point (section 7. Policy implications) of the article needs not to be isolated in an independent section but can be included in the discussion one or even in the conclusions.
Reply to Point 3: Thank you for your valuable suggestion. We have removed section 7 and included the policy implications in section 6, together with the concluding remarks. Please find these changes in the revised version of the manuscript.
Point 4: The numbering of the mathematical expressions of the econometric models should be sequential beginning in (1), (2)... The authors cannot repeat the same number with a different expression (for instance in 2.3.3). Not all the expressions are numbered. Several expressions with different numbers are exactly the same.
Reply to Point 4: Thank you for your suggestion. We have renumbered all mathematical expressions in a sequential manner. Please find these changes in the revised version of the manuscript.
Point 5: There are expressions that are not needed. There are models that are not previously introduced and explained (f. ex, the one of the table 6.).
Reply to Point 5: Thank you for your suggestion. We believe that all expressions included in the current version are important for conveying our empirical results. We have explained all variables used in the model from Table 6. Please find these changes in the revised version of the manuscript.
Point 6: The interpretation of the results is very poor and have many limitations. Some interpretations of the descriptive statistics are not needed and some of them are not well defined.
Reply to Point 6: Thank you for your observation. We have carefully checked and reworded the interpretation of the results. Moreover, we have discarded some paragraphs referring to the descriptive statistics, which were not necessary in the paper. Please find these changes in the revised version of the manuscript.
Point 7: There are still some errors and omissions in the English language.
Reply to Point 7: We have carefully proofread the English language used in this revised version and we believe the current wording is in accordance with the journal requirements and high academic standards of empirical research. In our opinion, we have followed the expected academic grammar standards in terms of tenses, phrasing, prepositions, idioms, etc. Moreover, we have formatted the paper and the reference list following the “Authors’ Guidelines”, including even the DOI for all sources (where applicable).
